# Estimating and interpreting nonlinear receptive field of sensory neural responses with deep neural network models

Menoua Keshishian[1,2], Hassan Akbari[1,2], Bahar Khalighinejad[1,2], Jose L Herrero[3,4], Ashesh D Mehta[3,4], Nima Mesgarani[1,2]*

[1]Department of Electrical Engineering, Columbia University, New York, United States; [2]Zuckerman Mind Brain Behavior Institute, Columbia University, New York, United States; [3]Feinstein Institute for Medical Research, Manhasset, United States; [4]Department of Neurosurgery, Hofstra-Northwell School of Medicine and Feinstein Institute for Medical Research, Manhasset, United States

**Abstract** Our understanding of nonlinear stimulus transformations by neural circuits is hindered by the lack of comprehensive yet interpretable computational modeling frameworks. Here, we propose a data-driven approach based on deep neural networks to directly model arbitrarily nonlinear stimulus-response mappings. Reformulating the exact function of a trained neural network as a collection of stimulus-dependent linear functions enables a locally linear receptive field interpretation of the neural network. Predicting the neural responses recorded invasively from the auditory cortex of neurosurgical patients as they listened to speech, this approach significantly improves the prediction accuracy of auditory cortical responses, particularly in nonprimary areas. Moreover, interpreting the functions learned by neural networks uncovered three distinct types of nonlinear transformations of speech that varied considerably from primary to nonprimary auditory regions. The ability of this framework to capture arbitrary stimulus-response mappings while maintaining model interpretability leads to a better understanding of cortical processing of sensory signals.

*For correspondence:
nima@ee.columbia.edu

**Competing interests:** The authors declare that no competing interests exist.

## Introduction

Creating computational models to predict neural responses from the sensory stimuli has been one of the central goals of sensory neuroscience research (*Hartline, 1940*; *Hubel and Wiesel, 1959*; *Hubel and Wiesel, 1962*; *Döving, 1966*; *Laurent and Davidowitz, 1994*; *Wilson, 2001*; *Boudreau, 1974*; *Mountcastle, 1957*). Computational models can be used to form testable hypotheses by predicting the neural response to arbitrary manipulations of a stimulus and can provide a way of explaining complex stimulus-response relationships. As such, computational models that provide an intuitive account of how sensory stimuli are encoded in the brain have been critical in discovering the representational and computational principles of sensory cortices (*Marr and Poggio, 1976*). One simple yet powerful example of such models is the linear receptive field model, which is commonly used in visual (*Hubel and Wiesel, 1959*; *Theunissen et al., 2001*) and auditory (*Theunissen et al., 2000*; *Klein et al., 2006*) neuroscience research. In the auditory domain, the linear spectrotemporal receptive field (STRF) (*Theunissen et al., 2001*; *Klein et al., 2006*; *Aertsen and Johannesma, 1981*) has led to the discovery of neural tuning to various acoustic dimensions, including frequency, response latency, and temporal and spectral modulation (*Miller et al., 2002*; *Woolley et al., 2005*; *Chi et al., 1999*). However, despite the success and ease of interpretability of

linear receptive field models, they lack the necessary computational capacity to account for the intrinsic nonlinearities of the sensory processing pathways (*David and Gallant, 2005*). This shortcoming is particularly problematic in higher cortical areas where stimulus representation becomes increasingly more nonlinear (*Christianson et al., 2008*; *Wu et al., 2006*). Several extensions have been proposed to address the limitations of linear models (*Paninski, 2004*; *Sharpee et al., 2004*; *Brenner et al., 2000*; *Kaardal et al., 2017*; *Ahrens et al., 2008*; *Mesgarani et al., 2009*; *Hong et al., 2008*; *Schwartz and Simoncelli, 2001*; *Schwartz et al., 2002*; *Butts et al., 2011*; *McFarland et al., 2013*; *Vintch et al., 2015*; *Harper et al., 2016*) (see (*Meyer et al., 2016*) for review). These extensions improve the prediction accuracy of neural responses, but this improvement comes at the cost of reduced interpretability of the underlying computation, hence limiting novel insights that can be gained regarding sensory cortical representation. In addition, these methods assume a specific model structure whose parameters are then fitted to the neural data. This assumed model architecture thus limits the range of the nonlinear transformations that they can account for. This lack of a comprehensive yet interpretable computational framework has hindered our ability to understand the nonlinear signal transformations that are found ubiquitously in the sensory processing pathways (*Hong et al., 2008*; *Abbott, 1997*; *Khalighinejad et al., 2019*).

A general nonlinear modeling framework that has seen great progress in recent years is the multi-layer (deep) neural network model (DNN) (*Hinton et al., 2006*; *LeCun et al., 2015*). Theses biologically inspired models are universal function approximators (*Hornik et al., 1989*) and can model any arbitrarily complex input-output relation. Moreover, these data-driven models can learn any form of nonlinearity directly from the data without any explicit assumption or prior knowledge of the nonlinearities. This property makes these models particularly suitable for studying the encoding properties of sensory stimuli in the nervous system (*Batty et al., 2016*; *McIntosh et al., 2016*; *Klindt et al., 2017*) whose anatomical and functional organization remains speculative particularly for natural stimuli. A major drawback of DNN models, however, is the difficulty in interpreting the computation that they implement because these models are analytically intractable (the so-called black box property) (*Mallat, 2016*). Thus, despite their success in increasing the accuracy of prediction in stimulus-response mapping, their utility in leading to novel insights into the computation of sensory nervous systems is lacking.

To overcome these challenges, we propose a nonlinear regression framework in which a DNN is used to model sensory receptive fields. An important component of our approach is a novel analysis method that allows for the calculation of the mathematically equivalent function of the trained neural network as a collection of stimulus-dependent, linearized receptive fields. As a result, the exact computation of the neural network model can be explained in a manner similar to that of the commonly used linear receptive field model, which enables direct comparison of the two models. Here, we demonstrate the efficacy of this nonlinear receptive field framework by applying it to neural responses recorded invasively in the human auditory cortex of neurosurgical patients as they listened to natural speech. We demonstrate that not only these models more accurately predict the auditory neural responses, but also uncover distinct nonlinear encoding properties of speech in primary and nonprimary auditory cortical areas. These findings show the critical need for more complete and interpretable encoding models of neural processing, which can lead to better understanding of cortical sensory processing.

## Results

### Neural recordings

To study the nonlinear receptive fields of auditory cortical responses, we used invasive electrocorticography (ECoG) to directly measure neural activity from five neurosurgical patients undergoing treatment for epilepsy. One patient had high-density subdural grid electrodes implanted on the left hemisphere, with coverage primarily over the superior temporal gyrus (STG). All five patients had depth electrodes (stereotactic EEG) with coverage of Heschl's gyrus (HG) and STG (*Figure 1A*). While HG and the STG are functionally heterogenous and each contain multiple auditory fields (*Nourski et al., 2014*; *Galaburda and Sanides, 1980*; *Morosan et al., 2001*; *Hickok and Saberi, 2012*; *Hamilton et al., 2018*), HG includes the primary auditory cortex, while the STG is considered mostly a nonprimary auditory cortical area (*Clarke and Architecture, 2012*). The patients listened to

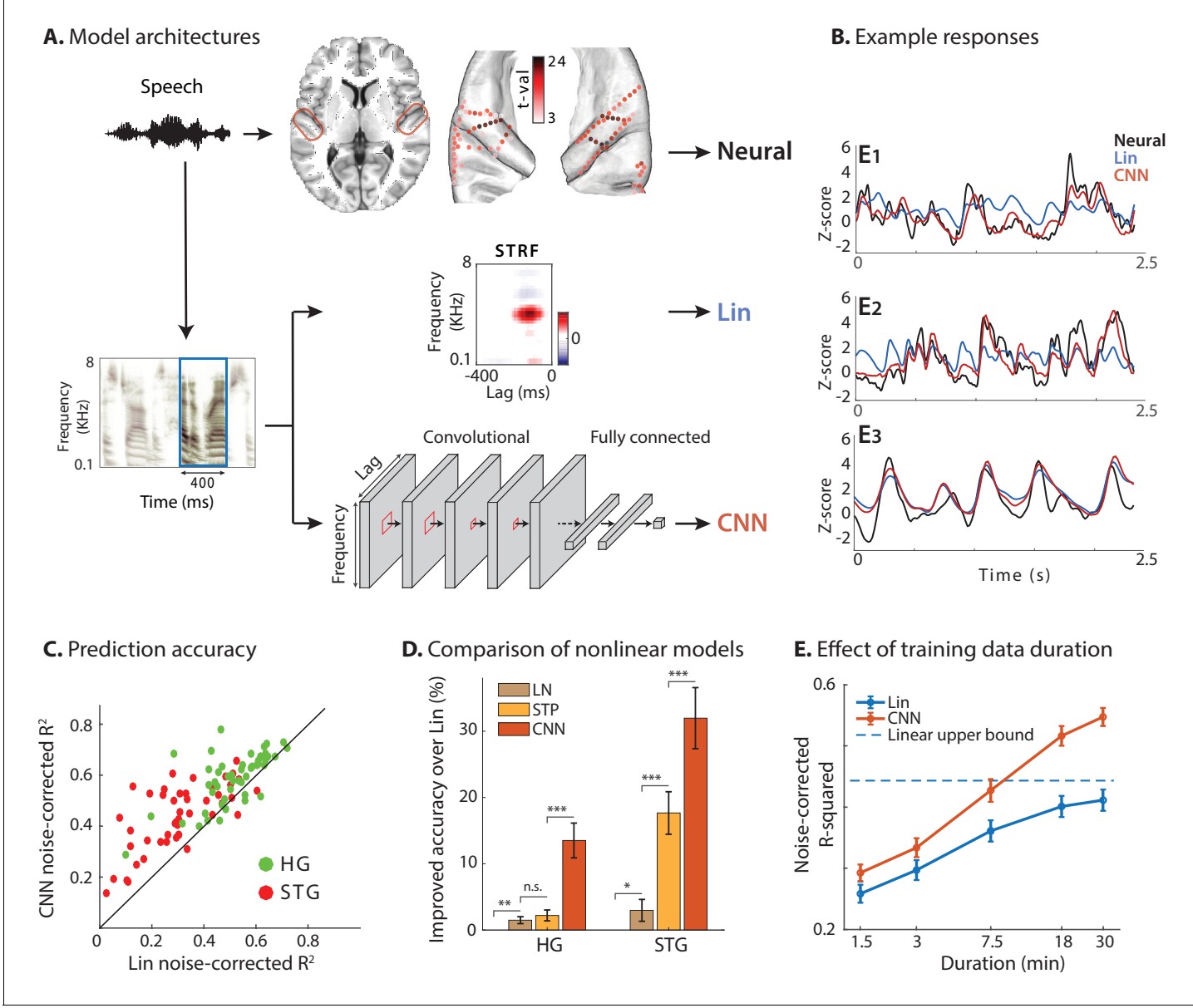

**Figure 1.** Predicting neural responses using linear and nonlinear regression models. (**A**) Neural responses to speech were recorded invasively from neurosurgical patients as they listened to speech. The brain plot shows electrode locations and t-value of the difference between the average response of a neural site to speech versus silence. The neural responses are predicted from the stimulus using a linear spectrotemporal receptive field model (Lin) and a nonlinear convolutional neural network model (CNN). Input to both models is a sliding time-frequency window with 400 ms duration. (**B**) Actual and predicted responses of three example sites using the Lin and CNN models. (**C**) Prediction accuracy of neural responses from the Lin and CNN models for sites in STG and HG. (**D**) Improved prediction accuracy over Lin model for linear-nonlinear (LN), short-term plasticity (STP), and CNN models. (**E**) Dependence of prediction accuracy on the duration of training data. Circles show average across electrodes and bars indicate standard error. The dashed line is the upper bound of average prediction accuracy for the Lin model across electrodes. The x-axis is in logarithmic scale. The online version of this article includes the following source data and figure supplement(s) for figure 1:

**Source data 1.** A MATLAB file containing four variables — group (location of electrode based on anatomy; 1 = Heschl's gyrus, 2 = superior temporal gyrus), rsquared_lin (noise-adjusted R-squared values of test set prediction by linear model), rqsuared_cnn (noise-adjusted R-squared values of test set prediction by CNN model), improvement (difference of the last two, as used in the *Figure 5B* prediction).
**Figure supplement 1.** Selecting stimulus window length for prediction.
**Figure supplement 2.** Hyperparameter optimization.

stories spoken by four speakers (two females) with a total duration of 30 min. All patients had self-reported normal hearing. To ensure that patients were engaged in the task, we intermittently paused the stories and asked the patients to repeat the last sentence before the pause. Eight separate sentences (40 s total) were used as the test data for evaluation of the encoding models, and each sentence was repeated six times in a random order.

We used the envelope of the high-gamma (70–150 Hz) band of the recorded signal as our neural response measure, which correlates with the neural firing in the proximity of electrodes (*Ray and Maunsell, 2011*; *Buzsáki et al., 2012*). The high-gamma envelope was extracted by first filtering neural signals with a bandpass filter and then calculating the Hilbert envelope. We restricted our analysis to speech-responsive electrodes, which were selected using a t-test between the average response of each electrode to speech stimuli versus the response in silence (t-value > 2). This criterion resulted in 50 out of 60 electrodes in HG and 47 out of 133 in STG. Electrode locations are plotted in *Figure 1A* on the average FreeSurfer brain (*Fischl et al., 2004*) where the colors indicate speech responsiveness (t-values).

We used the auditory spectrogram of speech utterances as the acoustic representation of the stimulus. Auditory spectrograms were calculated using a model of the peripheral auditory system that estimates a time-frequency representation of the acoustic signal on a tonotopic axis (*Yang et al., 1992*). The speech materials were split into three subsets for fitting the models – training, validation, and test. Repetitions of the test set were used to compute a noise-corrected performance metric to reduce the effect of neural noise on model comparisons. The remainder of the data was split between training and validation subsets (97% and 3%, respectively). There was no stimulus overlap between the three subsets. We used the prediction accuracy on the validation set to choose the best weights for the model throughout training.

## Linear and nonlinear encoding models

For each neural site, we fit one linear (Lin) and one nonlinear (CNN) regression model to predict its activity from the auditory spectrogram of the stimulus (*Figure 1A*). The input to the regression models was a sliding window with a duration of 400 ms with 10 ms steps, which was found by maximizing the prediction accuracy (*Figure 1—figure supplement 1*). The linear encoding model was a conventional STRF that calculates the linear mapping from stimulus spectrotemporal features to the neural response (*Theunissen et al., 2001*). The nonlinear regression model was implemented using a deep convolutional neural network (CNN; *LeCun et al., 1998*) consisting of two stages: a feature extraction network and a feature summation network. This commonly used nonlinear regression framework (*LeCun et al., 1990*; *Krizhevsky et al., 2012*; *Pinto et al., 2009*) consists of extracting a high-dimensional representation of the input (feature extraction) followed by a feature summation network to predict neural responses. The feature extraction network comprises three convolutional layers each with eight 3 × 3 2D convolutional kernels, followed by two convolutional kernels with 1 × 1 kernels to reduce the dimensionality of the representation, thus decreasing the number of model parameters. The feature summation stage was a two-layer fully connected network with 32 nodes in the hidden layer and a single output node. All hyperparameters of the network were determined by optimizing the prediction accuracy (*Figure 1—figure supplement 2*). All hidden layers had rectified linear unit (ReLU) nonlinearity (*Nair and Hinton, 2010*), and the output node was linear. A combination of mean-squared error and Pearson correlation was used as the training loss function (see Materials and methods).

## Predicting neural responses using linear and nonlinear encoding models

Examples of actual and predicted neural responses from the Lin and CNN models are shown in *Figure 1B* for three sample electrodes. We examined the nonlinearity of each neural site by comparing the prediction accuracy of Lin and CNN models. As *Figure 1B* shows, the CNN predictions (red) are more similar to the actual responses (black) compared to Lin predictions (blue). This observation confirms that the CNN model can capture the variations in the neural responses to speech stimuli more accurately. To quantify this improvement across all sites, we calculated the noise-adjusted R-squared value (*Dean et al., 2005*) (see Materials and methods) between the predicted and actual neural responses for each model. The scatter plot in *Figure 1C* shows the comparison of these values obtained for Lin and CNN models for each electrode, where the electrodes are colored by their

respective brain region. *Figure 1C* shows higher accuracy for CNN predictions compared to Lin predictions for the majority of electrodes (87 out of 97; avg. difference: 0.10; p<<1, t-test). Even though the overall prediction accuracy is higher for HG electrodes than ones in STG, the absolute improved prediction of CNN over Lin is significantly higher for STG electrodes (avg. improvement 0.07 in HG, 0.13 in STG; p<0.003, two sample one-tailed t-test). This higher improvement in STG electrodes reveals a larger degree of nonlinearity in the encoding of speech in the STG compared to HG.

To compare the CNN model with other common nonlinear extensions of the linear model, we predicted the neural responses using linear-nonlinear (LN) and short-term plasticity (STP) models (*Abbott, 1997*; *David et al., 2009*; *David and Shamma, 2013*; *Tsodyks et al., 1998*; *Lopez Espejo et al., 2019*). *Figure 1D* shows the improvement of each model over the linear model, averaged separately across electrodes in HG and STG. While all models improve the prediction accuracy significantly compared to the linear model, the CNN accuracy is considerably higher than the other two, particularly in higher auditory areas (STG) which presumably contain more nonlinearities.

Since models with more free parameters are more difficult to fit to limited data, we also examined the effect of training data duration on prediction accuracy for both Lin and CNN models (*Figure 1E*, see Materials and methods). While we can calculate an upper bound for achievable noise-corrected R-squared for the linear model (*David and Gallant, 2005*), deep neural networks are universal approximators (*Hornik et al., 1989*) and hence do not have a theoretical upper bound. Assuming a logarithmic relationship between prediction error $(1 - \rho^2)$ and amount of training data, our data suggests that doubling the amount of training data reduces CNN's prediction error on average by 10.8% ± 0.5% (standard error). *Figure 1E* also shows that the CNN predicts the neural responses significantly better than the Lin model even if the duration of the training data is short. The consistent superiority of the CNN model shows the efficacy of the regularization methods which help avoid the problem of local minima during the training phase.

## Interpreting the nonlinear receptive field learned by CNNs

The previous analysis demonstrates the superior ability of the CNN model to predict the cortical representation of speech particularly in higher order areas, but it does not show what types of nonlinear computation result in improved prediction accuracy. To explain the mapping learned by the CNN model, we developed an analysis framework that finds the mathematical equivalent linear function that the neural network applies to each instance of the stimulus. This equivalent function is found by estimating the derivative of the network output with respect to its input (i.e. the data Jacobian matrix [*Wang et al., 2016*]). We refer to this linearized equivalent function as the dynamic spectrotemporal receptive field (DSTRF). The DSTRF can be considered an STRF whose spectrotemporal tuning depends on every instance of the stimulus. As a result, the linear weighting function of the CNN model that is applied to each stimulus instance can be visualized in a manner similar to that of the STRF (see *Videos 1* and *2*).

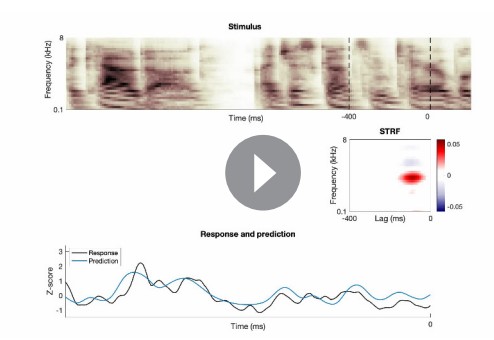

**Video 1.** The spectrotemporal receptive field (STRF) model applies a weight function to the stimulus to predict the neural response.

https://elifesciences.org/articles/53445#video1

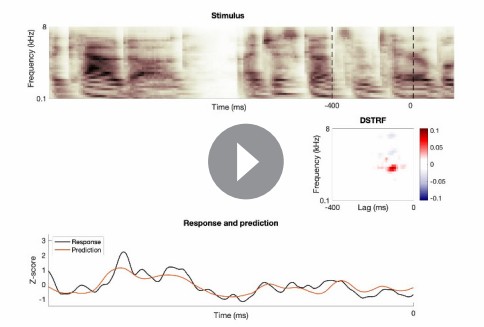

**Video 2.** The dynamic spectrotemporal receptive field (DSTRF) applies a time-varying, stimulus-dependent weight function to the stimulus to predict the neural response.

https://elifesciences.org/articles/53445#video2

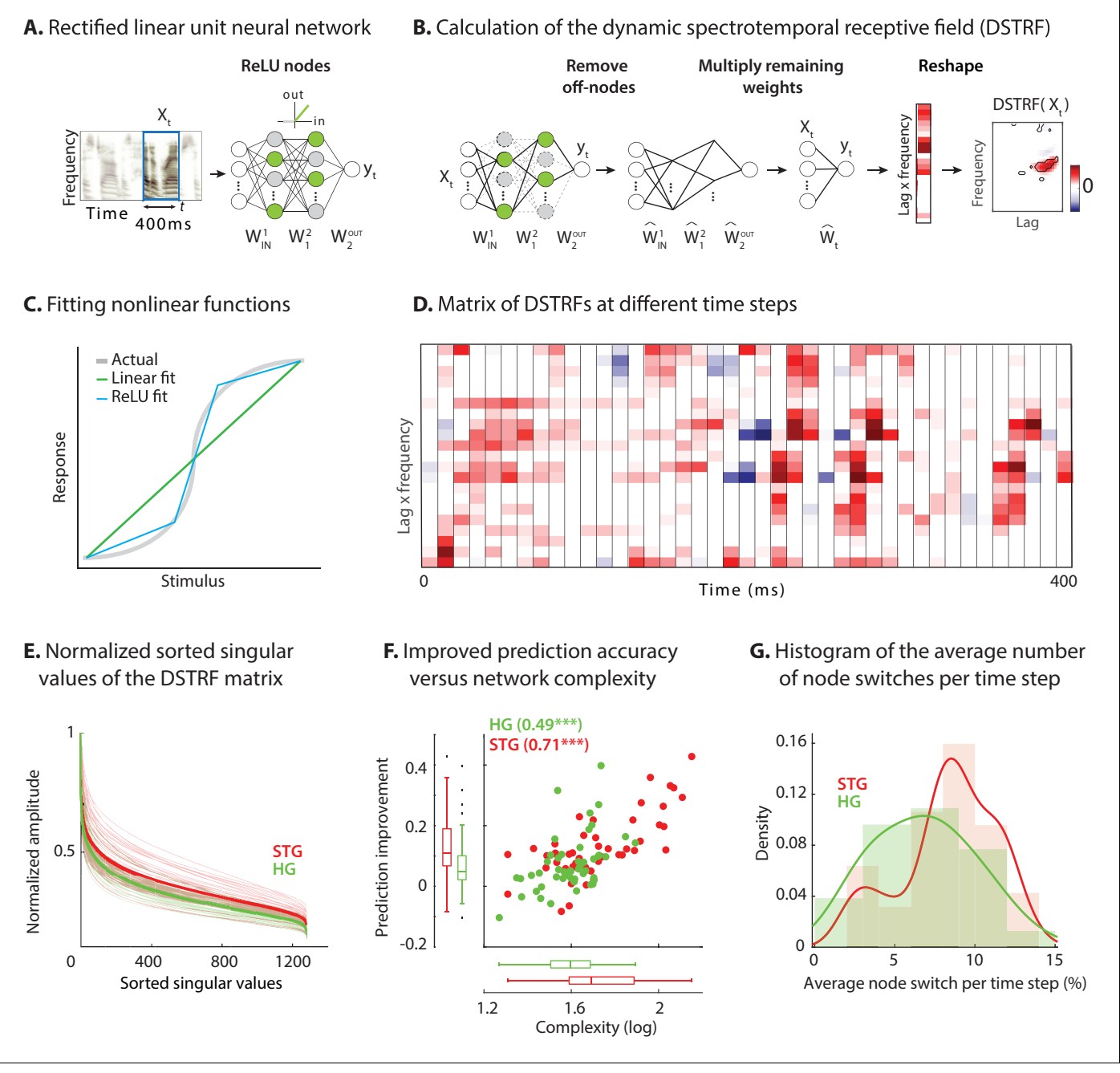

**Figure 2.** Calculating the stimulus-dependent dynamic spectrotemporal receptive field. (**A**) Activation of nodes in a neural network with rectified linear (ReLU) nonlinearity for the stimulus at time $t$. (**B**) Calculating the stimulus-dependent dynamic spectrotemporal receptive field (DSTRF) for input instance $X_t$ by first removing all inactive nodes from the network and replacing the active nodes with the identity function. The DSTRF is then computed by multiplying the remaining network weights. Reshaping the resulting weight matrix expresses the DSTRF in the same dimensions as the input stimulus and can be interpreted as a multiplicative template applied to the input. Contours indicate 95% significance (jackknife). (**C**) Comparison of piecewise linear (rectified linear neural network) and linear (STRF) approximations of a nonlinear function. (**D**) DSTRF vectors (columns) shown for 40 samples of the stimulus. Only a limited number of lags and frequencies are shown at each time step to assist visualization. (**E**) Normalized sorted singular values of the DSTRF matrix show higher diversity of the learned linear function in STG sites than in HG sites. The bold lines are the averages of sites in the STG and in HG. The complexity of a network is defined as the sum of the sorted normalized singular values. (**F**) Comparison of network complexity and the improved prediction accuracy over the linear model in STG and HG areas. (**G**) Histogram of the average number of switches from on/off to off/on states at each time step for the neural sites in the STG and HG.

The online version of this article includes the following source data and figure supplement(s) for figure 2:

**Source data 1.** A MATLAB file containing one variable — complexity (neural network complexity, as used in *Figure 5A* prediction).

**Figure supplement 1.** Converting convolution to matrix multiplication.

**Figure supplement 2.** DSTRF robustness across initializations.

**Figure supplement 3.** Comparing DSTRFs to STRFs.

Finding the DSTRF is particularly straightforward for a neural network with rectified linear unit nodes (ReLU), because ReLU networks implement piecewise linear functions (*Figure 2C*). A rectified linear node is inactive when the sum of its inputs is negative or is active and behaves like a linear function when the sum of its inputs is positive (*Figure 2A*). Obtaining a linear equivalent of a CNN for a given stimulus consists of first removing the weights that connect to the inactive nodes in the network (*Figure 2B*) and replacing the active nodes with identity functions. Next, the remaining weights of each layer are multiplied to calculate the overall linear weighting function applied to the stimulus. Because the resulting weighting vector has the same dimension as the input to the network, it can be visualized as a multiplicative template similar to an STRF (*Figure 2B*). The mathematical derivation of DSTRF is shown in Materials and methods (see also *Figure 2—figure supplement 1*).

To assign significance to the lag-frequency coefficients of the DSTRFs, we used the jackknife method (*Efron, 1982*) to fit multiple CNNs (n = 20) using different segments of the training data. Each of the 20 CNNs were trained by systematically excluding 1/20 of the training data. As a result, the response to each stimulus sample in the test data can be predicted from each of the 20 CNNs, resulting in 20 DSTRFs. The jackknife method results in a distribution for each lag-frequency coefficient of the DSTRF. The resulting DSTRF is the average of this distribution, where the variance signifies the uncertainty of each coefficient. We denoted the significance of the DSTRF coefficients by requiring all positive or all negative values for at least 19 out of the 20 models (corresponding to p = %95, jackknife) as shown by contours in *Figure 2B*. We used this method to mask the insignificant coefficients of the DSTRF (p>0.05, jackknife) by setting them to zero.

An example of DSTRF at different time points for one electrode is shown in *Figure 2D* where each column is the vectorized DSTRF (time lag by frequency) that is applied to the stimulus at that time point (for better visibility only part of the actual matrix is shown). This matrix contains all the variations of the receptive fields that the network applies to the stimulus at different time points. On one extreme, a network could apply a fixed receptive field to the stimulus at all times (similar to the linear model) for which the columns of the matrix in *Figure 2D* will all be identical. At the other extreme, a network can learn a unique function for each instance of the stimulus for which the columns of the matrix in *Figure 2D* will all be different functions. Because a more nonlinear function results in a higher number of piecewise linear regions (*Pascanu et al., 2014*; *Figure 2C*), the diversity of the functions in the lag-frequency by time matrix (columns in *Figure 2D*) indicates the degree of nonlinearity of the function that the network has learned. To quantify this network nonlinearity, we used singular-value decomposition (*Strang, 1993*) of the lag-frequency by time matrix. Each singular value indicates the variance in its corresponding dimension; therefore, the steepness of the sorted singular values is inversely proportional to the diversity of the learned functions. The sorted normalized singular values for all electrodes in HG and STG are shown in *Figure 2E*, demonstrating that the neural network models learn considerably more diverse functions for STG electrodes (statistical analysis provided below). This result confirms the increase in nonlinearity observed earlier in STG electrodes compared to HG electrodes.

## Complexity of the nonlinear receptive field

We defined the complexity of the CNN model as the sum of the normalized singular values of the lag-frequency by time matrix (*Figure 2D*). The complexity for all electrodes in HG and STG is shown in *Figure 2F* and is compared against the improved prediction accuracy of the CNN over the linear model. *Figure 2F* shows a significantly higher complexity for STG electrodes than for HG electrodes (HG avg.: 41.0, STG avg.: 59.7; p<0.001, two sample t-test), which also correlates with the prediction improvement of each electrode over the linear model (r = 0.66, p<0.001). Alternatively, a separate metric to measure the degree of network nonlinearity is the average number of nodes that switch between active and inactive states at each time step. The histogram of the average switches for HG

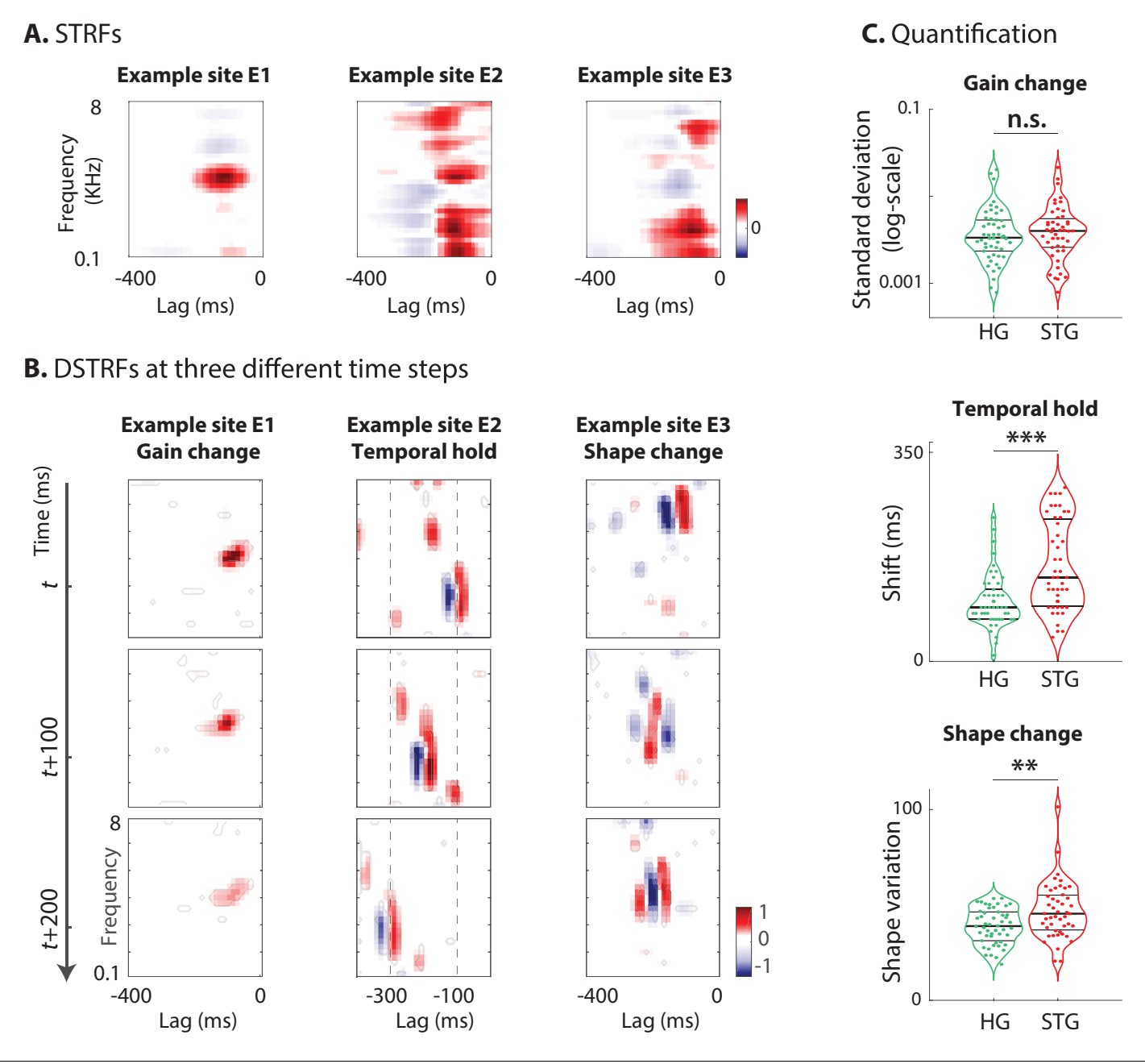

**Figure 3.** Characterizing the types of DSTRF variations. Three types of DSTRF variations are shown over time for three example sites that exhibit each of these types more prominently. The DSTRFs have been masked by 95% significance per jackknifing ($n = 20$). (**A**) The STRF of the three example sites. (**B**) Example site E1: Gain change of DSTRF, shown as the time-varying magnitude of the DSTRF at three different time points. Although the overall shape of the spectrotemporal receptive field is the same, its magnitude varies across time. E2: Temporal hold property of an DSTRF, seen as the tuning of this site to a fixed spectrotemporal feature but with shifted latency (lag) in consecutive time frames. E3: Shape change property of DSTRF, seen as the change in the spectrotemporal tuning pattern for different instances of the stimulus. (**C**) From top to bottom, distribution of gain change, temporal hold, and shape change values for sites in HG and STG. Horizontal lines mark quartiles. Temporal hold and shape change show significantly higher values in the STG.

The online version of this article includes the following source data and figure supplement(s) for figure 3:

**Source data 1.** A MATLAB file containing three variables, one for each type of quantified nonlinearity — nonlin_gain_change (gain change nonlinearity), nonlin_temporal_hold (temporal hold nonlinearity), nonlin_shape_change (shape change nonlinearity).

**Figure supplement 1.** Calculating temporal hold.

**Figure supplement 2.** Nonlinearity robustness to initialization and data.

**Video 3.** Visualization of gain change, temporal hold, and shape change for three example electrodes, each demonstrating one type of nonlinearity more prominently.
https://elifesciences.org/articles/53445#video3

and STG electrodes shows significantly higher values for STG electrodes (*Figure 2G*; p<0.01, one-tailed two sample t-test). This observation validates the finding that the larger improvement of prediction accuracy in STG is due to the implementation of a more diverse set of linear templates, whereas the HG electrodes require a smaller number of linear functions to accurately capture the stimulus response relationships in this area. Importantly, these results are not dependent on network parameters and network initialization. While the hyperparameters and the training of the network can change the internal implementation of the input-output function, the function itself is robust and remains unchanged (Materials and methods and *Figure 2—figure supplement 2*). Furthermore, the linearized functions averaged over all samples closely resemble the STRF for the corresponding electrode, with the similarity decreasing with the complexity of the neural site (*Figure 2—figure supplement 3*).

## Identifying and quantifying various types of nonlinear receptive field properties

We showed that the nonlinear function of the CNN model can be expressed as a collection of linearized functions. To investigate the properties of these linearized functions learned by the models for various neural sites, we visually inspected the DSTRFs and observed three general types of variations over time, which we refer to as: I) gain change, II) temporal hold, and III) shape change. We describe and quantify each of these three nonlinear computations in this section using three example electrodes that exhibit each of these types more prominently (*Video 3*). The STRFs for the three examples sites are shown in *Figure 3A*.

### Gain change nonlinearity

The first and simplest type of DSTRF change is gain change, which refers to the time-varying magnitude of the DSTRF. This effect is shown for one example site (E1) at three different time points in *Figure 3B*. Although the overall shape of the spectrotemporal filter applied to the stimulus at these three time points is not different, the magnitude of the function varies considerably over time. This variation in the gain of the stimulus-response function that is learned by the CNN may reflect the nonlinear adaptation of neural responses to the acoustic stimulus (*Abbott, 1997*; *Rabinowitz et al., 2011*; *Mesgarani et al., 2014a*). We quantified the degree of gain change for each site by calculating the standard deviation of the DSTRF magnitude over the stimulus duration.

### Temporal hold nonlinearity

The second type of DSTRF change is temporal hold, which refers to the tuning of a site to a fixed spectrotemporal feature but with shifted latency in consecutive time frames (*Figure 3B*). This particular tuning nonlinearity results in a sustained response to arbitrarily fast spectrotemporal features, hence resembling a short-term memory of that feature. An example of temporal hold for a site (E2) is shown in *Figure 3B* where the three plots show the DSTRF of this site at three consecutive time steps. Even though the overall shape of the spectrotemporal feature tuning remains the same, the latency (lag) of the feature shifts over time with the stimulus. This nonlinear property decouples the duration of the response to an acoustic feature from the temporal resolution of that feature. For example, temporal hold could allow a network to model a slow response to fast acoustic features. This computation cannot be done with a linear operation because a linear increase in the analysis time scale inevitably results in the loss of temporal resolution, as shown in the STRF of site E2 in *Figure 3A*.

We quantified the temporal hold for each site by measuring the similarity of consecutive DSTRFs once the temporal shift is removed. This calculation assumes that when there is temporal hold nonlinearity, the $DSTRF_t(\tau, f)$ is most correlated with $DSTRF_{t+n}(\tau - n, f)$. To quantify temporal hold, we compared the correlation values for all time points $1 \leq t \leq T$ in two conditions: when the consecutive

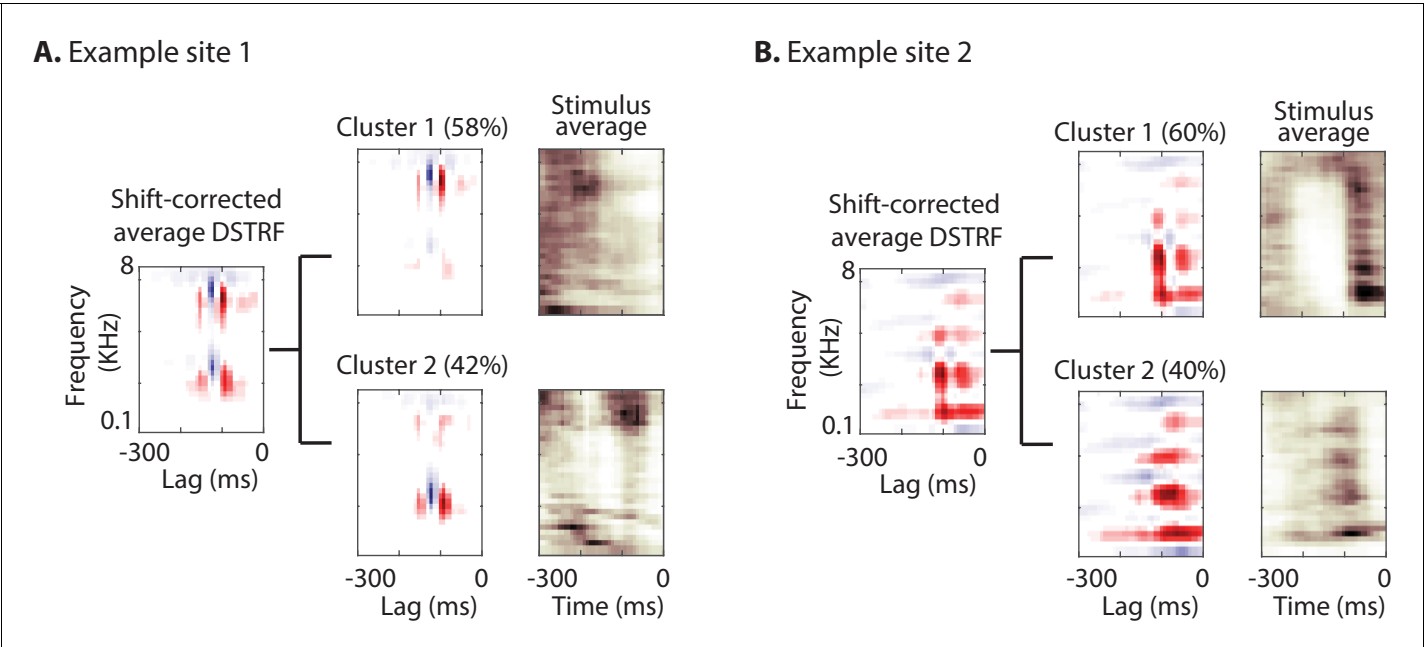

**Figure 4.** Characterizing the spectrotemporal tuning of electrodes with clustering. (A, B) The average shift-corrected DSTRF for two example sites and average k-means clustered DSTRFs based on the similarity of DSTRFs (correlation distance). Each cluster shows tuning to a distinct spectrotemporal feature. The average DSTRF shows the sum of these different features. The average spectrograms over the time points at which each cluster DSTRF was selected by the network demonstrate the distinct time-frequency pattern in the stimuli that caused the network to choose the corresponding template.

DSTRF is shifted by $n$ lags and when it is not shifted. For each $n$, the two conditions are compared using a one-tailed Wilcoxon signed-rank test (Materials and methods). The largest $n$ for which there is a significant positive change between the shit and no-shift conditions ($p < 0.05$) quantifies the temporal hold for that electrode (*Figure 3—figure supplement 1*). This calculation assumes that the duration of temporal hold does not depend on instances of the stimulus. An interesting path for future inquiry would be to study the dependence of this effect on specific stimuli.

## Shape change nonlinearity

The last dimension of DSTRF variation is shape change, which refers to a change in the spectrotemporal tuning of a site across stimuli. Intuitively, a model that implements a more nonlinear function will have a larger number of piecewise linear regions, each exhibiting a different DSTRF shape. An example of shape change for a site (E3) is shown in *Figure 3B*, showing three different spectrotemporal patterns at these three different time points. To quantify the degree of change in the shape of the DSTRF for a site, we repeated the calculation of network complexity but after removing the effect of temporal hold from the DSTRFs (see Materials and methods). Therefore, shape change is defined as the sum of the normalized singular values of the shift-corrected lag-frequency by time matrix (*Figure 2D*). Thus, the shape change indicates the remaining complexity of the DSTRF function that is not due to temporal hold, or gain change (aligned samples differing only in amplitude are captured by the same set of eigenvectors). This stimulus-dependent change in the spectrotemporal tuning of sites reflects a nonlinearity that appears as the sum of all possible shapes in the STRF, as shown in *Figure 3B*.

The distribution of the three nonlinearity types defined here across all neural sites in HG and STG areas are shown in *Figure 3C*. Looking at these distributions can give us a better understanding of the shared nonlinear properties among neural populations of each region. The degree of gain change for electrodes in HG and STG areas spans a wide range. However, we did not observe a significant difference between the gain change values in STG and HG sites (HG avg.: 4.6e-3, STG avg.: 4.9e-3; p=0.23, Wilcoxon rank-sum), suggesting a similar degree of adaptive response in HG and STG sites. The distribution of temporal hold values for all sites in the STG and HG shows significantly larger values in STG sites (HG avg.: 10.1, STG avg.: 15.7; p<1e-4, Wilcoxon rank-sum). This increased

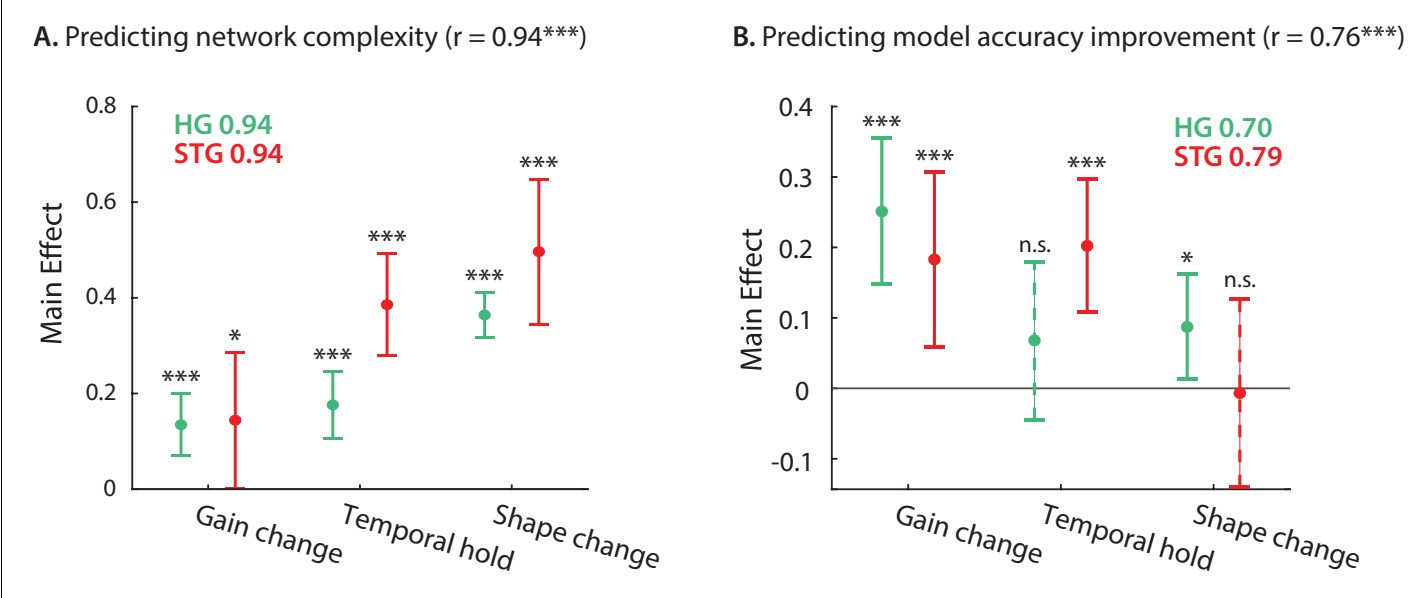

**Figure 5.** Contribution of DSTRF variations to network complexity and prediction improvement. (**A**) Predicting the complexity of the neural networks from gain change, temporal hold, and shape change using a linear regression model. The main effect of regression analysis shows the significant contribution of all nonlinear parameters in predicting the network complexity in both STG and in HG sites. Legend shows prediction R-values separately for each region. (**B**) Predicting the improved accuracy of neural networks over the linear model from the three nonlinear parameters for each site in the STG and HG. The main effect of the regression analysis shows different contribution of the three nonlinear parameters in predicting the improved accuracy over the linear model in different auditory cortical areas. Figure legends shows prediction R-values separately for each region.
The online version of this article includes the following figure supplement(s) for figure 5:

**Figure supplement 1.** Complexity and accuracy improvement predicted values.
**Figure supplement 2.** Covarying nonlinear parameters.

temporal hold is consistent with the previous findings showing increased temporal integration in higher auditory cortical areas (*King and Nelken, 2009*; *Berezutskaya et al., 2017*), which allows the stimulus information to be combined across longer time scales. The shape change values are significantly higher on average in STG sites than in HG (HG avg.: 38.8, STG avg.: 46.8; p=0.005, Wilcoxon rank-sum), demonstrating that STG sites have more nonlinear encoding, which requires more diverse receptive field templates than HG sites. Finally, similar to DSTRF shape, these quantified nonlinearity parameters were highly consistent across different network initializations and different segments of the test data (*Figure 3—figure supplement 2*).

## Finding subtypes of receptive fields

As explained in the shape change nonlinearity, the nonlinear model may learn several subtypes of receptive fields for a neural site. To further investigate the subtypes of receptive fields that the CNN model learns for each site, we used the k-means algorithm (*Lloyd, 1982*) to cluster shift-corrected DSTRFs based on their correlation similarity. The optimal number of clusters for each site was determined using the gap statistic method (*Tibshirani et al., 2001*). The optimal number of clusters across all sites differed from 1 to 6; the majority of sites, however, contained only one main cluster (84.9% of sites; mean number of clusters = 1.43 ± 1.26 SD). *Figure 4* shows the DSTRF clustering analysis for two example sites, where for each cluster the average DSTRF and the average auditory stimulus for the time points that have DSTRFs belonging to that cluster are shown. The average DSTRFs in *Figure 4* show two distinct receptive field shapes that the CNN models apply to the stimulus at different time points. In addition, the average spectrograms demonstrate the distinct time-frequency power in the stimuli that caused the model to choose the corresponding template.

## Contribution of nonlinear variations to network complexity and prediction improvement

So far, we have defined and quantified three types of nonlinear computation for each neural site – gain change, temporal hold, and shape change – resulting in three numbers describing the stimulus-response nonlinearity of the corresponding neural population. Next, we examined how much of the network complexity (*Figure 2F*) and improved accuracy over a linear model (*Figure 1D*) can be accounted for by these three parameters. We used linear regression to calculate the complexity and prediction improvement for each site from the gain change, temporal hold, and shape change parameters (*Figure 5*). The predicted and actual complexity of the models are shown in *Figure 5—figure supplement 1*. The high correlation value (r = 0.94, p < 1e-41) confirms the efficacy of these three parameters to characterize the complexity of the stimulus-response mapping across sites. Moreover, the main effects of the regression (*Seber and Lee, 2012*) shown in *Figure 5A* suggest a significant contribution from all three parameters to the overall complexity of DSTRFs in both HG and STG. The high correlation values between the actual and predicted improved accuracy (*Figure 5—figure supplement 1*; r = 0.76, p < 1e-16) show that these three parameters also largely predict stimulus-response nonlinearity. The contribution of each nonlinear factor in predicting the improved prediction, however, is different between the HG and STG areas, as shown by the main effects of the regression in *Figure 5B*. The gain change (effect = 0.183, p = 0.005) and temporal hold (effect = 0.202, p = 1e-4) factors contributed to the improved accuracy in STG sites, while the gain change is the main predictor of the improvement in HG sites (effect = 0.250, p = 2e-5), with a modest contribution from shape change (effect = 0.087, p = 0.028). This result partly reiterates the encoding distinctions we observed in HG and STG sites where temporal hold and shape change were significantly higher in STG than HG (*Figure 3*). Notably, the three types of DSTRF variations are not independent of each other and covary considerably (*Figure 5—figure supplement 2*). This interdependence may explain why the shape change parameter significantly predicts the improved prediction accuracy in HG, but not in STG, because shape change covaries considerably more with the temporal hold parameter in STG (correlation between shape change and temporal hold $r = 0.24$, $p = 0.09$ in HG; $r = 0.57$, p < 1e - 4 in STG). In other words, sites in STG appear to exhibit multiple nonlinearities simultaneously, making it more difficult to discern their individual contributions. Together, these results demonstrate how our proposed nonlinear encoding method can lead to a comprehensive, intuitive way of studying nonlinear mechanisms in sensory neural processing by utilizing deep neural networks.

## Discussion

We propose a general nonlinear regression framework to model and interpret any complex stimulus-response mapping in sensory neural responses. This data-driven method provides a unified framework that can discover and model various types of nonlinear transformations without needing any prior assumption about the nature of the nonlinearities. In addition, the function learned by the network can be interpreted as a collection of linear receptive fields from which the network chooses for different instances of the stimulus. We demonstrated how this method can be applied to auditory cortical responses in humans to discover novel types of nonlinear transformation of speech signals, therefore extending our knowledge of the nonlinear cortical computations beyond what can be explained with previous models. While the unexplained noise-corrected variance by a linear model indicates the overall nonlinearity of a neural code, this quantity alone is generic and does not inform the types of nonlinear computations that are being used. In contrast, our proposed method unravels various types of nonlinear computation that are present in the neural responses and provides a qualitative and quantitative account of the underlying nonlinearity.

Together, our results showed three distinct nonlinear properties which largely account for the complexity of the neural network function and could predict the improved prediction accuracy over the linear model. Extracting these nonlinear components and incorporating them in simpler models such as the STRF can systematically test the contribution and interaction of these nonlinear aspects more carefully. However, such simplification and abstraction proved nontrivial in our data because the different types of variation we describe in this paper are not independent of each other and covary considerably. Interestingly, the prediction accuracy improvement of the STP model was significant only in STG responses. The improved prediction accuracy using STP model was also correlated

with the gain change parameter (partial $r = 0.24$, $p = 0.018$, Spearman, controlling for temporal hold and shape change) and not temporal hold or shape change (p > 0.2 for partial correlations with improvement), meaning that sites with larger gain change saw a greater increase from the STP model. Our STP models included four recovery time constant $\tau$ and four release probability $u$ parameters. We performed a simplified comparison using the average $\tau$ and $u$ for each electrode with our parameters. A partial correlation analysis of $u$, which is an indicator of plasticity strength, revealed only a positive correlation with gain change (partial $r = 0.28$, $p = 0.006$, Spearman, controlling for temporal hold and shape change; p > 0.8 for partial correlation of $u$ and temporal hold and shape change). On the other hand, $\tau$ had a negative correlation with temporal hold (partial $r = -0.22$, $p = 0.032$, Spearman, controlling for gain change and shape change; p > 0.16 for partial correlation with gain change; p > 0.7 for partial correlation with shape change). These findings are in line with our hypothesis that the gain change nonlinearity captures the nonlinear adaptation of the neural responses to the short-term history of the stimulus.

The increasing nonlinearity of the stimulus-response mapping throughout the sensory pathways (*King and Nelken, 2009*; *Chechik et al., 2006*) highlights the critical need for nonlinear computational models of sensory neural processing, particularly in higher cortical areas. These important nonlinear transformations include nonmonotonic and nonlinear stimulus-response functions (*Sadagopan and Wang, 2009*), time-varying response properties such as stimulus adaptation and gain control (*Abbott, 1997*; *Rabinowitz et al., 2011*; *Mesgarani et al., 2014a*; *Dean et al., 2008*), and nonlinear interaction of stimulus dimensions and time-varying stimulus encoding (*Machens et al., 2004*). These nonlinear effects are instrumental in creating robust perception, which requires the formation of invariant perceptual categories from a highly variable stimulus (*Leaver and Rauschecker, 2010*; *Chang et al., 2010*; *de Heer et al., 2017*; *Bidelman et al., 2013*; *Mesgarani et al., 2014b*; *Steinschneider, 2013*; *Russ et al., 2007*). The previous research on extending simple receptive field models that have tried to address the linear system limitations include generalized linear models (*Paninski, 2004*), linear-nonlinear (LN) models (*Sharpee et al., 2004*; *Brenner et al., 2000*; *Kaardal et al., 2017*), input nonlinearity models (*Ahrens et al., 2008*; *David et al., 2009*), gain control models (*Hong et al., 2008*; *Schwartz and Simoncelli, 2001*; *Schwartz et al., 2002*), context-dependent encoding models, and LNLN cascade models (*Butts et al., 2011*; *McFarland et al., 2013*; *Vintch et al., 2015*; *Harper et al., 2016*) (see (*Meyer et al., 2016*) for review). Even though all these models improve the prediction accuracy of neural responses, this improvement comes at the cost of reduced interpretability of the computation. For example, the multifilter extensions of the auditory STRF (*Sharpee et al., 2004*; *Brenner et al., 2000*; *Kaardal et al., 2017*; *Butts et al., 2011*; *McFarland et al., 2013*; *Vintch et al., 2015*; *Harper et al., 2016*) lead to nonlinear interactions of multiple linear models, which is considerably less interpretable than the STRF model. Our approach extends the previous methods by significantly improving the prediction accuracy over the linear, linear-nonlinear, and STP models while at the same time, remaining highly interpretable.

The computational framework we propose to explain the neural network function (*Nagamine and Mesgarani, 2017*) can be used in any feedforward neural network model with any number of layers and nodes, such as in fully connected networks, locally connected networks (*Coates and Ay, 2011*), or CNNs (*LeCun and Bengio, 1995*). Nonetheless, one limitation of the DSTRF method is that it cannot be used for neural networks with recurrent connections. Because feedforward models use a fixed duration of the signal as the input, the range of the temporal dependencies and contextual effects that can be captured with these models is limited. Nevertheless, sensory signals such as speech have long-range temporal dependencies for which recurrent networks may provide a better fit. Although we found only a small difference between the prediction accuracy of feedforward and recurrent neural networks in our data (about 1% improvement in HG, 3% in STG), the recent extensions of the feedforward architecture, such as dilated convolution (*Luo and Mesgarani, 2018*) or temporal convolutional networks (*Lea et al., 2016*), can implement receptive fields that extend over long durations. Our proposed DSTRF method would seamlessly generalize to these architectures, which can serve as an alternative to recurrent neural networks when modeling the long-term dependencies of the stimulus is crucial. Furthermore, while we trained a separate model for each electrode, it is possible to use a network with shared parameters to predict all neural sites. This direction can also be used to examine the connectivity and specialization of the representation across various regions (*Kell et al., 2018*).

In summary, our proposed framework combines two desired properties of a computational sensory-response model; the ability to capture arbitrary stimulus-response mappings and maintaining model interpretability. We showed that this data-driven method reveals novel nonlinear properties of cortical representation of speech in the human brain which provides an example for how it can be used to create more complete neurophysiological models of sensory processing in the brain.

# Materials and methods

## Participants and neural recordings

Five patients with pharmacoresistant focal epilepsy were included in this study. All patients underwent chronic intracranial encephalography (iEEG) monitoring at Northshore University Hospital to identify epileptogenic foci in the brain for later removal. Four patients were implanted with stereo-electroencephalographic (sEEG) depth arrays only and one with both depth electrodes and a high-density grid (PMT, Chanhassen, MN). Electrodes showing any sign of abnormal epileptiform discharges, as identified in the epileptologists' clinical reports, were excluded from the analysis. All included iEEG time series were manually inspected for signal quality and were free from interictal spikes. All research protocols were approved and monitored by the institutional review board at the Feinstein Institute for Medical Research (IRB-AAAD5482), and informed written consent to participate in research studies was obtained from each patient before electrode implantation. A minimum of 45 electrodes per brain area was determined to be sufficient for our between region significance comparison analyses.

iEEG signals were acquired continuously at 3 kHz per channel (16-bit precision, range ±8 mV, DC) with a data acquisition module (Tucker-Davis Technologies, Alachua, FL). Either subdural or skull electrodes were used as references, as dictated by recording quality at the bedside after online visualization of the spectrogram of the signal. Speech signals were recorded simultaneously with the iEEG for subsequent offline analysis. The envelope of the high-gamma response (70–150 Hz) was extracted by first filtering neural signals with a bandpass filter and then using the Hilbert transform to calculate the envelope. The high-gamma responses were z-scored and resampled to 100 Hz.

## Brain maps

Electrode positions were mapped to brain anatomy using registration of the postimplant computed tomography (CT) to the preimplant MRI via the postop MRI. After coregistration, electrodes were identified on the postimplantation CT scan using BioImage Suite. Following coregistration, subdural grid and strip electrodes were snapped to the closest point on the reconstructed brain surface of the preimplantation MRI. We used FreeSurfer automated cortical parcellation (*Fischl et al., 2004*) to identify the anatomical regions in which each electrode contact was located with a resolution of approximately 3 mm (the maximum parcellation error of a given electrode to a parcellated area was <5 voxels/mm). We used Destrieux parcellation, which provides higher specificity in the ventral and lateral aspects of the medial lobe. Automated parcellation results for each electrode were closely inspected by a neurosurgeon using the patient's coregistered postimplant MRI.

## Stimulus

Speech materials consisted of continuous speech stories spoken by four speakers (two male and two female). The duration of the stimulus was approximately 30 minutes and was sampled at 11025 Hz. The data was split into two segments for training (30 min) and validation (50 s). Additionally, eight sentences totaling 40 s were used for testing the model and presented to the patients six times to improve the signal-to-noise ratio. There was no overlap between the training, test, and validation sets. The input to the regression models was a sliding window of 400 ms (40 timesteps), which was chosen to optimize prediction accuracy (*Figure 1—figure supplement 1*). The windowing stride was set to one to maintain the same final sampling rate, and as a result, the two consecutive input vectors to the regression models overlapped at 39 time points.

## Acoustic representation

An auditory spectrogram representation of speech was calculated from a model of the peripheral auditory system (*Yang et al., 1992*). This model consists of the following stages: (1) a cochlear filter

bank consisting of 128 constant-Q filters equally spaced on a logarithmic axis, (2) a hair cell stage consisting of a low-pass filter and a nonlinear compression function, and (3) a lateral inhibitory network consisting of a first-order derivative along the spectral axis. Finally, the envelope of each frequency band was calculated to obtain a time-frequency representation simulating the pattern of activity on the auditory nerve. The final spectrogram has a sampling frequency of 100 Hz. The spectral dimension was downsampled from 128 frequency channels to 32 channels to reduce the number of model parameters.

## Calculating spectrotemporal receptive fields (STRFs)

Linear STRF models were fitted using the STRFlab MATLAB toolbox (*Theunissen et al., 2001*; *STRFlab, 2020*). For each electrode, a causal model was trained to predict the neural response at each time point from the past 400 ms of stimulus. The optimal model sparsity and regularization parameters were chosen by maximizing the mutual information between the actual and predicted responses for each electrode.

## Extensions of the linear model

We used the Neural Encoding Model System (NEMS) python library (*David, 2018*) to fit both linear-nonlinear (LN) and short-term plasticity (STP) models (*Abbott, 1997*; *David et al., 2009*; *David and Shamma, 2013*), using its TensorFlow backend. For the LN model, we used a rank-4 time-frequency separable model with four gaussian kernels for selecting frequency bands, a 400 ms (40 sample) finite impulse response (FIR) filter for each selected frequency band, followed by a double exponential static nonlinearity with trainable parameters:

$$\sigma(x) = b + a * exp(-exp(-e^{\kappa} * (x - c)))$$
(1)

We chose this nonlinearity because it performed better than the ReLU in our data. Each gaussian kernel is applied to the input spectrogram separately to create a four-channel output. Each channel is then convolved with its corresponding FIR filter and the final nonlinearity is applied to the sum of the output channels.

For the STP model, we added a short-term plasticity module to the setup above, between the frequency selection kernels and the temporal response filters. The STP module is parameterized by the Tsodyks-Markram model (*Tsodyks et al., 1998*; *Lopez Espejo et al., 2019*) and has two parameters for each of the four selected frequency bands: release probability $u$ determining the strength of plasticity, and time constant $\tau$ determining the speed of recovery. The corresponding STP equations for channel $i$ are as follows, where $x_i$ and $y_i$ are the input and output to the module and $d_i$ is the change in gain due to plasticity:

$$d_i(t) = d_i(t-1) + x_i(t-1)[1 - d_i(t-1)]u_i - \frac{d_i(t-1)}{\tau_i}$$
(2)

$$y_i(t) = d_i(t)x_i(t)$$
(3)

## DNN architecture

We designed a two-stage DNN consisting of feature extraction and feature summation modules (*Figure 1A*). In this framework, a high-dimensional representation of the input is first calculated (feature extraction network), and this representation is then used to regress the output of the model (feature summation network). The feature extraction stage consists of three convolutional layers with eight 3x3 2D convolutional kernels each, followed by a convolutional layer with four 1x1 kernels to reduce the dimensionality of the representation. The output of this stage is the input to another convolutional layer with a single 1x1 kernel. A 1x1 kernel that is applied to an input with $N$ channels has 1x1xN parameters, and its output is a linear combination of the input channels, thus reducing the dimension of the latent variables and, consequently, the number of trainable parameters. The feature summation stage is a two-layer fully connected network with a hidden layer of 32 nodes, followed by an output layer with a single node. All layers except the output layer have $L2$ regularization, dropout (*Srivastava et al., 2014*), ReLU (*Nair and Hinton, 2010*) activations, and no bias. The output layer has regularization, linear activation, and a bias term. The parameters of the

model, including the number of layers, the size of the convolutional kernels, and the number of fully connected nodes, were found by optimizing the prediction accuracy (*Figure 1—figure supplement 2*).

## DNN training and cross-validation

The networks were implemented in Keras using the TensorFlow backend. A separate network was trained for each electrode. Kernel weight initializations were performed using a method specifically developed for DNNs with rectified linear nonlinearities (*He et al., 2015*) for faster convergence. We used ReLU nonlinearities for all layers except the last layer, and dropouts with $p = 0.3$ for the convolutional layers and $p = 0.4$ for the first fully connected layer were used to maximize prediction accuracy. The convolutional layers had strides of one, and their inputs were padded with zeros such that the layer's output would have the same dimension as the input. We applied an L2 penalty (Ridge) with regularization constant of 0.001 to the weights of all the layers. Each training epoch had a batch size of 128, and optimization was performed using Adam with an initial learning rate of 0.0001. Networks were trained with a maximum of 30 epochs, with early stopping when the validation loss did not decrease for five consecutive epochs. The weights that resulted in the best validation loss during all training epochs were chosen as the final weights. The loss function was a linear combination of the MSE and Pearson's correlation coefficient:

$$\frac{1}{n}\sum_i (y_i - \hat{y}_i)^2 - \frac{\sum_i (y_i - \bar{y}_i)(\hat{y}_i - \bar{\hat{y}}_i)}{\sqrt{\sum_i (y_i - \bar{y}_i)^2 \sum_i (\hat{y}_i - \bar{\hat{y}}_i)^2}} \tag{4}$$

in which $y$ is the high-gamma envelope of the recorded neural data from a given electrode, and $\hat{y}$ is the predicted response of the neural network. We chose this loss because it outperformed the MSE loss in our data.

## Evaluating model performance (noise-corrected correlation)

To account for the variations in neural responses that are not due to the acoustic stimulus, we repeated the test stimulus six times to more accurately measure the explainable variance. To obtain a better measure of the model's goodness of fit, we used a noise-corrected R-squared value instead of the simple correlation. Having n responses to the same stimulus, $r_1$ to $r_n$, we defined $R_o$ and $R_e$ as the averages of odd and even numbered trials. Then, we calculated the noise-corrected correlation according to the following equations, where $\rho_c^2$ is our reported R-squared of the noise-corrected Pearson correlation. Assume that $R_e$ and $R_o$ consist of the same true signal ($\hat{R}$) with variance $\sigma_{\hat{R}}^2$ and i. i.d. gaussian noise ($n_e$, $n_o$) with variance $\sigma_n^2$. Given $R_e$, $R_o$, and model prediction $P$, we want to find the correlation between $P$ and the true signal $\hat{R}$ (*Schoppe et al., 2016*).

$$R_e = \hat{R} + n_e, R_o = \hat{R} + n_o, \sigma_{R_e} = \sigma_{R_o} = \sigma_R = \sqrt{\sigma_{\hat{R}}^2 + \sigma_n^2} \tag{5}$$

$$\rho_{P,R_o} = \frac{COV(P, R_o)}{\sigma_P \sigma_{R_o}} = \frac{COV(P, \hat{R}) + COV(P, n_o)}{\sigma_P \sigma_R} \approx \frac{COV(P, \hat{R})}{\sigma_P \sigma_R} \approx \rho_{P,R_e} \tag{6}$$

$$\rho_{R_o,R_e} = \frac{COV(R_o, R_e)}{\sigma_{R_o} \sigma_{R_e}} = \frac{COV(\hat{R}, \hat{R}) + COV(\hat{R}, n_e) + COV(n_o, \hat{R}) + COV(n_o, n_e)}{\sigma_R^2} \approx \frac{\sigma_{\hat{R}}^2}{\sigma_R^2} \tag{7}$$

$$\rho_c = \frac{\frac{1}{2}(\rho_{P,R_e} + \rho_{P,R_o})}{\sqrt{\rho_{R_o,R_e}}} = \frac{COV(P, \hat{R})}{\sigma_P \sigma_R} \frac{\sigma_R}{\sigma_{\hat{R}}} = \frac{COV(P, \hat{R})}{\sigma_P \sigma_{\hat{R}}} = \rho_{P,\hat{R}} \tag{8}$$

## Computing DSTRFs for convolutional neural networks

The first step for calculating the dynamic spectrotemporal receptive field (DSTRF) of a CNN consists of converting the CNN into a multilayer perceptron (MLP) (*Figure 2—figure supplement 1*) because calculating DSTRFs for an MLP is more straightforward. To achieve this task, we must first convert each convolutional layer to its equivalent locally connected layer, which is essentially a sparse fully connected layer. To do so, we find the equivalent matrix $W$ for convolutional kernels $K_1 - K_l$, where

$K_i$ is the $i$-th kernel of the convolutional layer. Transforming all layers of the CNN into fully connected layers results in an MLP network. The input and output tensors of the fully connected layers have only a single dimension, which is usually not the case for convolutional layers. Hence, the inputs and outputs of all layers in the equivalent network are the flattened versions of the original network.

Assume that all zeros tensor $W$ has dimensions $M \times N \times C \times M \times N \times L$ where $M$ and $N$ are, respectively, the rows and columns of the input to a convolutional layer; $C$ is the number of channels in the input; and $L$ is the number of kernels in a layer. Additionally, assume $K_l$ (the $l$-th kernel of the layer) has dimensions $H \times W \times C$, where H and W are the height and width of the kernel, respectively, and $C$ is defined as before. We begin by populating $W$ according to *Equation 9* for all values of $m$, $n$, and $l$. Then, we reshape $W$ to $(M * N * C) \times (M * N * L)$ to obtain the 2D matrix that will transform the flattened input of the convolutional layer to the flattened output. Of course, $W$ can be directly populated as a 2D matrix for improved performance.

$$W\left[m - \left\lfloor \frac{H}{2} \right\rfloor : m + \left\lfloor \frac{H}{2} \right\rfloor, n - \left\lfloor \frac{W}{2} \right\rfloor : n + \left\lfloor \frac{W}{2} \right\rfloor, 1:C, m, n, l\right] = K_l \tag{9}$$

The calculation of DSTRFs for the equivalent MLP network involves few steps (*Figure 2B*). The DSTRF of a network with ReLU activations and no bias in the intermediate layers is equivalent to the gradient of the network's output with respect to the input vector (*Nagamine and Mesgarani, 2017*):

$$DSTRF_\theta(x_t) = \frac{\partial \hat{y}_t}{\partial x_t} = \frac{\partial \hat{y}_t}{\partial z_t^l} \frac{\partial z_t^l}{\partial h_t^{l-1}} \frac{\partial h_t^{l-1}}{\partial z_t^{l-1}} \frac{\partial z_t^{l-1}}{\partial h_t^{l-2}} \cdots \frac{\partial h_t^1}{\partial z_t^1} \frac{\partial z_t^1}{\partial x_t} \tag{10}$$

$$= \frac{\partial \hat{y}_t}{\partial z_t^l} W_{l-1}^l \frac{\partial h_t^{l-1}}{\partial z_t^{l-1}} W_{l-2}^{l-1} \cdots \frac{\partial h_t^1}{\partial z_t^1} W_{IN}^1 \tag{11}$$

where $z_t^l$ is the weighted sum of inputs to nodes in layer $l$ for input $x_t$, $h_t^l$ represents the output of nodes in layer $l$ to the same input, and $\theta$ denotes the dependence on the parameters of the network. In a network with ReLU nodes:

$$\frac{\partial h(\cdot)}{\partial z(\cdot)} = \begin{cases} 1 & if \ z > 0 \\ 0 & if \ z < 0 \end{cases} \tag{12}$$

which means that we can replace the product of $\frac{\partial h(\cdot)}{\partial z(\cdot)}$ and $W_{l-1}^l$ with an adjusted weight matrix $\hat{W}_{l-1}^l$, where $m$ and $n$ are indices of nodes in layer $l$ and $l-1$:

$$\hat{W}_{l-1}^l(x_t)[m,n] = \begin{cases} W_{l-1}^l[m,n] & if \ h_t^l[m] > 0 \\ 0 & otherwise \end{cases} \tag{13}$$

$$DSTRF(x_t) = \hat{W}_{l-1}^l \hat{W}_{l-2}^{l-1} \ldots \hat{W}_{IN}^1 \tag{14}$$

Because DSTRF can be defined as the gradient of the output with respect to the input of the network, we can avoid the manual calculation process by utilizing TensorFlow's built-in automatic differentiation capability.

## Inference of statistical bounds on DSTRF coefficients

To estimate the statistical confidence on the values of DSTRFs, we used the jackknife method (*Efron, 1982*). We partitioned the full training data into 20 segments with roughly the same length. We then removed one segment at a time and fit a model using the remaining 19 segments. This procedure results in 20 total trained models for each electrode. During the prediction phase, we calculate the DSTRF from each model for a given stimulus, which results in a distribution for each lag-frequency value. The resulting DSTRF is the mean of this distribution, and the standard error for each coefficient is calculated using the jackknife formula:

$$\hat{\theta} = \frac{1}{n} \sum_{i=1}^n \hat{\theta}_i \ , \quad SE_{jack} = \sqrt{\frac{n-1}{n} \sum_{i=1}^n \left(\hat{\theta}_i - \hat{\theta}\right)^2}, \tag{15}$$

where $\hat{\theta}_i$ is the estimate of the coefficient when removing the $i$-th block of the data, $\hat{\theta}$ is the average of the $n$ estimates, and $SE_{jack}$ is the standard error estimate. To assign significance to the coefficients, we found the lag-frequency coefficients that were all positive or all negative for at least %95 of the models (19 out of 20). We denote patches of significant coefficients with black contours.

## Complexity estimation

To quantify the nonlinearity of the network receptive field, we measure the diversity of the equivalent linear functions that the network learns for different instances of the stimulus. To measure this function diversity, we calculated the singular-value decomposition (*Strang, 1993*) of the matrix containing all the linearized equivalent functions of a network (*Figure 2D*). Each singular value indicates the variance in its corresponding dimension; therefore, the steepness of the sorted singular values is inversely proportional to the diversity of the functions that are learned by the network. We define the complexity of the network as the sum of the normalized singular values where $\sigma_i$ is the $i$-th element of the singular values vector, and $D$ is the length of the vector:

$$complexity = \frac{1}{\max_i \sigma_i} \sum_{i=1}^{D} \sigma_i \tag{16}$$

## Estimation of gain change

We calculated the magnitude of the DSTRF at each time point using its standard deviation (*equation 17*, F = 32, T = 40). The gain change parameter for each site was then defined as the standard deviation of the DSTRF magnitude over the duration of the test stimulus. This quantity measures the degree to which the magnitude of the DSTRF changes across different instances of the stimuli.

$$|DSTRF_t| = \sqrt{\frac{\sum_{f=1}^{F} \sum_{\tau=1}^{T} \left(DSTRF_t[f, \tau] - \overline{DSTRF_t}\right)^2}{F * T - 1}} \tag{17}$$

$$gain\ change = \sqrt{\frac{\sum_{t=1}^{N} \left(|DSTRF_t| - |\overline{DSTRF}|\right)^2}{N - 1}} \tag{18}$$

## Estimation of temporal hold

Calculation of the temporal hold parameter for a given recording site involves three steps. For a given DSTRF at time $t$, $DSTRF_t(\tau, f)$, we first calculate its correlation with $DSTRF_{t+n}(\tau, f)$ and its shift-corrected version, $DSTRF_{t+n}(\tau - n, f)$:

$$\alpha_t(n) = corr(DSTRF_t(\tau, f)\ ,\ DSTRF_{t+n}(\tau, f))\ , \qquad 1 \le n \le 30 \tag{19}$$

$$\beta_t(n) = corr(DSTRF_t(\tau, f)\ ,\ DSTRF_{t+n}(\tau - n, f))\ , \quad 1 \le n \le 30 \tag{20}$$

The upper limit 30 corresponds to 300 ms and was empirically found to be sufficient for the range of temporal hold seen in our data. Next, for each $n$, we perform a one-tailed Wilcoxon signed-rank test to determine if there is a significant positive change between $\beta_t(n)$ and $\alpha_t(n)$ pairs across the entire test set ($t = 1 \ldots T$). Finally, the temporal hold is defined as the largest $n$ for which the test yields a significant result (p < 0.05). Intuitively, these steps find the largest duration in time for which a spectrotemporal pattern persists, assuming that the latency of that pattern shifts over time to keep it aligned with a specific feature in the stimulus (see *Figure 3—figure supplement 1* for examples).

## Estimation of shape change

The shape change parameter represents the diversity of the linear functions that is not due to the gain change or temporal hold nonlinearities. To calculate this parameter, we first removed the effect of temporal hold nonlinearity by time-aligning the DSTRF instances to the average DSTRF across the entire stimuli. This operation is done by finding the best shift, $n_t$, for DSTRFs at each time instant,

$DSTRF_t(\tau - n_t, f)$. The values of $n_t$ are found iteratively by maximizing the correlation between $DSTRF_t(\tau - n_t, f)$ and the average DSTRF over the entire stimulus duration: $\overline{DSTRF}(\tau, f) = \frac{1}{T}\sum_{t=1}^{T} DSTRF_t(\tau, f)$. At the end of each iteration, the average DSTRF is updated using the new shifted DSTRFs, and this operation is repeated until $n_t$ values converge, or a maximum number of iterations is reached. In our data, $n_t$ converged within fifty iterations. After removing the temporal hold effect, we repeated the same procedure used for the calculation of network complexity (*Equation 16*) but instead, used the time-aligned DSTRFs to perform the singular-value decomposition. Aligned DSTRFs with same spectrotemporal features but different gain values are captured by the same eigenvectors. The sum of the sorted normalized singular values indicates the diversity of the linear functions learned by the network due to a change in their spectrotemporal feature tuning.

## Dependence of DSTRFs on stimulus and initialization

We used neural networks as our nonlinear encoding model which are fitted using stochastic gradient descent algorithms. Because reaching the global minimum in such optimization methods cannot be guaranteed, it is possible that our results may depend on the initialization of network parameters prior to training and the stochasticity involved in training. We tested the robustness of the DSTRF shapes using $n = 10$ different random initializations of the network weights (*He et al., 2015*). For each electrode, we split the network instances into two groups of $n/2$, and compared the average $DSTRF_t(\tau, f)$ across the two groups. We also looked at the relation between the amplitude of an DSTRF and its consistency (*Figure 2—figure supplement 1*).

We did a similar analysis for the nonlinearity values extracted from the network, comparing the parameters extracted from the average DSTRFs of each group. In addition, the DSTRFs are estimated by inputting the held-out data (test set) into the neural networks. As a result, the DSTRFs are inherently stimulus dependent. To study the effect of limited test data on DSTRF shapes and the nonlinearity values, we repeated our analysis twice by using non-overlapping halves of the test data. Results are shown in *Figure 3—figure supplement 2*.

## Code availability

The codes for pre-processing the ECoG signals and calculating the high-gamma envelope are available at http://naplab.ee.columbia.edu/naplib.html (*Khalighinejad et al., 2017*). Python codes for training the CNN encoding models and computing the DSTRFs, and MATLAB functions for calculating the nonlinearity parameters (gain change, temporal hold, and shape change), are available at https://github.com/naplab/DSTRF (*Keshishian, 2020*; copy archived at https://github.com/elifes-ciences-publications/DSTRF). The videos and a link to the Git repository are also available on the project website: http://naplab.ee.columbia.edu/dstrf.html.

## Acknowledgements

This work was funded by a grant from the National Institutes of Health, NIDCD-DC014279, and the National Science Foundation CAREER award.

## Additional information

### Funding

| Funder | Grant reference number | Author |
|---|---|---|
| National Institutes of Health | NIDCD-DC014279 | Menoua Keshishian<br>Hassan Akbari<br>Bahar Khalighinejad<br>Nima Mesgarani |
| National Institute of Mental Health | | Jose L Herrero<br>Ashesh D Mehta |
| National Science Foundation | National Science Foundation CAREER awards | Menoua Keshishian<br>Nima Mesgarani |

The funders had no role in study design, data collection and interpretation, or the decision to submit the work for publication.

## Author contributions
Menoua Keshishian, Software, Formal analysis, Validation, Investigation, Visualization, Methodology, Writing - original draft, Writing - review and editing; Hassan Akbari, Software, Formal analysis, Investigation, Methodology; Bahar Khalighinejad, Data curation, Formal analysis; Jose L Herrero, Resources, Data curation; Ashesh D Mehta, Resources, Data curation, Investigation; Nima Mesgarani, Conceptualization, Resources, Data curation, Formal analysis, Supervision, Funding acquisition, Validation, Investigation, Visualization, Methodology, Writing - original draft, Project administration, Writing - review and editing

## Author ORCIDs
Menoua Keshishian (iD) https://orcid.org/0000-0003-0368-288X
Ashesh D Mehta (iD) http://orcid.org/0000-0001-7293-1101
Nima Mesgarani (iD) https://orcid.org/0000-0002-2987-759X

## Ethics
Human subjects: All research protocols were approved and monitored by the institutional review board at the Feinstein Institute for Medical Research (IRB-AAAD5482), and informed written consent to participate in research studies was obtained from each patient before electrode implantation.

## Decision letter and Author response
Decision letter https://doi.org/10.7554/eLife.53445.sa1
Author response https://doi.org/10.7554/eLife.53445.sa2

# Additional files
## Supplementary files
• Transparent reporting form

## Data availability
Source data files have been provided for Figures 1-3. Raw data cannot be shared as we do not have ethical approval to share this data. To request access to the data, please contact the corresponding author.

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
