## [Decision Letter]

**Acceptance summary:**

This manuscript provides a novel take on modeling cortical responses using general, nonlinear neural network models while maintaining a high and desirable level of interpretability. The quantitative results demonstrate the success of the method, in comparison to more classic approaches, and the subsequent analysis provides an accessible and insightful interpretation of the “strategies” used by the neural network models to achieve this improved performance.

**Decision letter after peer review:**

Thank you for submitting your article "Estimating and interpreting nonlinear receptive fields of sensory responses with deep neural network models" for consideration by *eLife*. Your article has been reviewed by two peer reviewers, and the evaluation has been overseen by a Reviewing Editor and Michael Frank as the Senior Editor. The following individuals involved in the review of your submission have agreed to reveal their identity: Bernhard Englitz (Reviewer #1); Nicholas A Lesica (Reviewer #2).

The reviewers have discussed the reviews with one another and the Reviewing Editor has drafted this decision to help you prepare a revised submission.

Summary:

The reviewers found that the manuscript provides a novel take on modeling cortical responses using general, nonlinear models while maintaining a high and desirable level of interpretability. In particular, the study demonstrates the potential of deep networks for the analysis of intracranial recordings from the human auditory cortex during the presentation of speech. The approach involves fitting deep networks and then analyzing linear equivalents of the fits to provide insights into nonlinear sensory processing. The reported quantitative results demonstrate the potential of the method, in comparison to simpler / classic models such as STRFs, and the subsequent analysis provides an accessible and insightful interpretation of the computational strategies used by the deep network models to achieve this improved performance.

As a general comment, the reviewers (and the reviewing editor) would like to encourage the authors to make software for their model publicly available.

Essential revisions:

The reviewers raised a number of concerns that must be adequately addressed before the paper can be accepted. Some of the required revisions will likely require further experimentation within the framework of the presented studies and techniques.

– Response evaluation with noise-adjusted correlation: The authors account for the possibility of noise in the responses by computing the noise-adjusted correlation, detailed in the Materials and methods. It was not clear to the reviewers where this method was referenced from since they could not find it mentioned in Dean, Harper and McApline, 2005 (though they could have easily missed it). The description in the Materials and methods is mathematically clear, but does not clarify the properties of this measure, e.g. under which circumstances would a 1 value be achieved? Please comment.

While this adjustment does something potentially related, one reviewer's suggestion is to use instead, the principled approach of predictive power (Sahani and Linden 2003, NeurIPS), which gives a good sense of absolute performance in the context of noise. Otherwise, the authors should explain/introduce the noise-adjusted correlation more fully.

– As presented, there is uncertainty about how much of what is being fitted is actually truly reflective of sensory processing vs. variability in the (linearized) fits due to the limited amount of data used for training. Since these networks are universal function approximators, they could in principle capture 100% of the explainable variance in the data if they are given enough training samples. In that perfectly predictive regime, we could be very confident that differences between fits reflect true differences in sensory processing. But if we are not in this regime, then we cannot know what fraction of the differences between fits is “real” without further investigation. The authors describe something along these lines in the section “Linearized function robustness”. But this only assesses robustness against different initial conditions for the fits, not the number of training samples. A more detailed evaluation of the robustness of the estimates as a function of the amount of available training data would help address this concern. Specifically, the reviewers would like to see some type of analysis that assesses the degree to which adding more data would increase performance and/or change the fits. Perhaps something along the lines of bootstrap resampling to at least put confidence intervals on a statistic would help.

– On a related note: since performance is measured as explainable variance (rather than just variance) the implicit goal is to capture only the stimulus-driven portion of the response. But the training is performed on single-trial data. With enough training samples the models will learn to ignore non-stimulus driven fluctuations even if they are trained on single-trial data, but how do we know that we have enough training samples to be in this regime? If we are not, then isn't there a worry that the fluctuations the time-point-by-time-point linearized fits reflect attempts to capture non-stimulus driven variance? In short, it is critical to quantify how much the (differences between) (linearized) fits would be expected to change if more training samples were used. Ideally, they wouldn't change at all. But given the reality of limited experimental data, we need a principled approach to derive confidence intervals just as we would for any other statistics, especially if this general approach is to be used meaningfully by non-experts.

– Comparison with more advanced neural models: While the currently presented method has the clear advantage of time-to-time interpretability of the kernel, the reviewers were not fully convinced that the absolute level of predictability was necessarily best-in-class (which is also not claimed in the text, but it is hard to assess where it falls within the set of existing models). It would thus be important to compare the DNN fit against two more general models, i.e. an STRF with a static nonlinearity and a model including adaptation (e.g., as proposed by Stephen David at OHSU). In particular, relating the extracted three properties to the adaptation parameter of the latter model could provide some unification between modeling approaches.

– Temporal hold: The identification of the temporal hold property was seen as one of the most exciting results from the manuscript. Hence, the reviewers request that it be analyzed more thoroughly. The criterion for the lag duration was just given as “significantly correlated”, without specification of a test, p-threshold, or a correction for multiple testing. In particular, it is not clear what happens if it drops briefly below the threshold but is significantly correlated afterward (e.g. oscillations?). Given the novelty of this result, it would be important to devote more detailed analysis to these questions (e.g., using permutation tests a la Koenig and Melie-Garcia, Brain Topography (2010), which is fast to implement and includes the corresponding options.)

– Method limitation: If one wants to measure the degree of nonlinearity in a given brain area, isn't it sufficient to simply look at how much of the explainable variance is or is not captured by a linear model? What else is learned by looking at the performance of a particular nonlinear model in this context? In fact, it would seem that using the linear model alone would be the only way to make a general statement in this context. Just because one particular nonlinear model fits one area better than another (or improves predictions in one area more than another) doesn't necessarily show that one or the other area is, in general, more nonlinear. It only suggests the nonlinearity in one area is more readily fit by that particular network given limited training samples. Please comment.

– Statistical testing: The statistical testing performed needs to be detailed more to include effect sizes.

---

## [Author Response]

Essential revisions:The reviewers raised a number of concerns that must be adequately addressed before the paper can be accepted. Some of the required revisions will likely require further experimentation within the framework of the presented studies and techniques.– Response evaluation with noise-adjusted correlation: The authors account for the possibility of noise in the responses by computing the noise-adjusted correlation, detailed in the Materials and methods. It was not clear to the reviewers where this method was referenced from since they could not find it mentioned in Dean, Harper and McApline, 2005 (though they could have easily missed it). The description in the Materials and methods is mathematically clear, but does not clarify the properties of this measure, e.g. under which circumstances would a 1 value be achieved? Please comment.While this adjustment does something potentially related, one reviewer's suggestion is to use instead, the principled approach of predictive power (Sahani and Linden 2003, NeurIPS), which gives a good sense of absolute performance in the context of noise. Otherwise, the authors should explain/introduce the noise-adjusted correlation more fully.

We have thoroughly expanded our derivation of the formula based on the normalized correlation measure (CCnorm) in (Schoppe et al., 2016) for N=2, and fixed the reference. Also, in that article Schoppe et al. relate the CCnorm to the signal power explained proposed by (Sahani and Linden, 2003).

“Having <inline-graphic mime-subtype="x-emf" mimetype="image" xlink:href="media/image1.emf" /> responses to the same stimulus, r1 to rn, we defined Ro and Re as the averages of odd and even numbered trials. Then, we calculated the noise-corrected correlation according to the following equations, where ρc2 is our reported R-squared of the noise-corrected Pearson correlation. Assume that Re and Ro consist of the same true signal (R^) with variance σR^2 and i.i.d. gaussian noise (ne, no) with variance σn2. Given Re, Ro, and model prediction P, we want to find the correlation between P and the true signal R^ (95).

Re=R^+ne, Ro=R^+no, σRe=σRo=σR=σR^2+σn2 (5)(6)ρP,Ro=COV(P,Ro)σPσRo=COV(P,R^)+COV(P,no)σPσR≈COV(P,R^)σPσR≈ρP,Re(7)ρRo,Re=COV(Ro,Re)σRoσRe=COV(R^,R^)+COV(R^,ne)+COV(no,R^)+COV(no,ne)σR2≈σR^2σR2

ρc=12(ρP,Re+ρP,Ro)ρRo,Re=COV(P,R^)σPσRσRσR^=COV(P,R^)σPσR^=ρP,R^ (8)”

By this notation, one would expect a value of 1 when the model can fully capture the true signal R^.

– As presented, there is uncertainty about how much of what is being fitted is actually truly reflective of sensory processing vs. variability in the (linearized) fits due to the limited amount of data used for training. Since these networks are universal function approximators, they could in principle capture 100% of the explainable variance in the data if they are given enough training samples. In that perfectly predictive regime, we could be very confident that differences between fits reflect true differences in sensory processing. But if we are not in this regime, then we cannot know what fraction of the differences between fits is “real” without further investigation. The authors describe something along these lines in the section “Linearized function robustness”. But this only assesses robustness against different initial conditions for the fits, not the number of training samples. A more detailed evaluation of the robustness of the estimates as a function of the amount of available training data would help address this concern. Specifically, the reviewers would like to see some type of analysis that assesses the degree to which adding more data would increase performance and/or change the fits. Perhaps something along the lines of bootstrap resampling to at least put confidence intervals on a statistic would help.

We agree with the reviewers that a better measure of uncertainty is needed. To address the question of how the performance of our fits would change with the amount of data, we have fit both the linear and CNN models to varying amounts of training data, specifically 20 random partitions of 5/10/25/60 percent (a la David and Gallant, 2005), in addition to using the full dataset 20 times. (Figure 1E)

For the relationship between the data length and performance for the linear model, we can use the formulation in (David and Gallant, 2005) to calculate an upper bound on the explained variance of the data by the model (dashed line in the figure shows average upper bound). Since neural networks are universal approximators, theoretically a deep neural network can reach the 100% noise-corrected explanatory power. Assuming added data has roughly a uniform diversity, and the amount of prediction error and data length follow a logarithmic relationship, we can estimate from our data that on average, the amount of noise-corrected unexplained variance will reduce by 10.8% by doubling the amount of training data.

To address the realness of the linearized fits, we have taken two precautions. First, our training, validation and test sets are disjoint, so that if a single trial noise is learned by the network, the chances of it showing up in the test set (where we do our linearized fit analysis) is minimal. Second, we have added a jackknife significance analysis to our computed linearized fits to determine their significant portions, which is explained in the answer to the next question:

– On a related note: since performance is measured as explainable variance (rather than just variance) the implicit goal is to capture only the stimulus-driven portion of the response. But the training is performed on single-trial data. With enough training samples the models will learn to ignore non-stimulus driven fluctuations even if they are trained on single-trial data, but how do we know that we have enough training samples to be in this regime? If we are not, then isn't there a worry that the fluctuations the time-point-by-time-point linearized fits reflect attempts to capture non-stimulus driven variance? In short, it is critical to quantify how much the (differences between) (linearized) fits would be expected to change if more training samples were used. Ideally, they wouldn't change at all. But given the reality of limited experimental data, we need a principled approach to derive confidence intervals just as we would for any other statistics, especially if this general approach is to be used meaningfully by non-experts.

This is an important point indeed. As noted in the previous section, we have added a jackknife analysis with n=20 to measure our confidence in the resulting linearized fits. To this end, we partitioned the training dataset into 20 equal-size segments, then trained 20 models by leaving out one segment at a time. For a given stimulus window, we then compute the linearized fits of each model. This analysis results in 20 values for the weights of each lag-frequency coefficient. We use the empirical mean of the distribution as our DSTRF weights and compute a standard error according to the jackknife formula, where θ^i is the estimated value from the model that was fit with the i-th block of data removed:

θ^=1n∑i=1nθ^i, SEjack=n−1n∑i=1n(θ^i−θ^)2, (15)

For a given lag-frequency coefficient in the DSTRF, if 95% (19/20) of these values are above zero, we denote that as a significantly positive weight, and if 95% are below zero, a significantly negative weight. We show these significant regions with contours in our update Figure 2.

In our data, almost all visible weights ended up as significant in this analysis. Hence, one can use these significance maps (corresponding to 95%) to mask the non-significant weights of the DSTRF for a given stimulus window, thus improving the signal to noise ratio of the estimated DSTRFs. The samples displayed in the new Figure 3 are masked by the significance maps.

Additionally, to obtain confidence intervals on the nonlinear parameter estimations extracted from the CNNs for each electrode, one can use a similar jackknifing analysis. Due to the significant computational cost involved in such an endeavor, we used a more practical analysis. We measured the robustness of our extracted parameters to training by computing them on two separate groups of trained models, and to test data by computing them on two splits of the test dataset. (Figure 3—figure supplement 2)

– Comparison with more advanced neural models: While the currently presented method has the clear advantage of time-to-time interpretability of the kernel, the reviewers were not fully convinced that the absolute level of predictability was necessarily best-in-class (which is also not claimed in the text, but it is hard to assess where it falls within the set of existing models). It would thus be important to compare the DNN fit against two more general models, i.e. an STRF with a static nonlinearity and a model including adaptation (e.g., as proposed by Stephen David at OHSU). In particular, relating the extracted three properties to the adaptation parameter of the latter model could provide some unification between modeling approaches.

We agree that comparison with other nonlinear models is important and necessary. To address this point, we fitted two additional models to our data, a linear model with static nonlinearity (LN), and a short-term plasticity (STP) model as implemented in Stephen David’s toolbox (Neural Encoding Model System, available at: https://github.com/LBHB/NEMS/). We compared the performance of the two additional models and our CNN with the baseline linear model (Figure 1D). The results indicate that the CNN significantly outperforms the other models, both in HG and STG (p < 0.001, paired t-test).

“Our STP models included four recovery time constant τ and four release probability u parameters. We performed a simplified comparison using the average τ and u for each electrode with our parameters. A partial correlation analysis of u, which is an indicator of plasticity strength, revealed only a positive correlation with gain change (partial r=0.28, p=0.006, Spearman, controlling for temporal hold and shape change; p>0.8 for partial correlation of u and temporal hold and shape change). On the other hand, τ had a negative correlation with temporal hold (partial r=−0.22, p=0.032, Spearman, controlling for gain change and shape change; p>0.16 for partial correlation with gain change; p>0.7 for partial correlation with shape change). These findings are in line with our hypothesis that the gain change nonlinearity captures the nonlinear adaptation of the neural responses to the short-term history of the stimulus.”

– Temporal hold: The identification of the temporal hold property was seen as one of the most exciting results from the manuscript. Hence, the reviewers request that it be analyzed more thoroughly. The criterion for the lag duration was just given as “significantly correlated”, without specification of a test, p-threshold, or a correction for multiple testing. In particular, it is not clear what happens if it drops briefly below the threshold but is significantly correlated afterward (e.g. oscillations?). Given the novelty of this result, it would be important to devote more detailed analysis to these questions (e.g., using permutation tests a la Koenig and Melie-Garcia, Brain Topography (2010), which is fast to implement and includes the corresponding options.)

We agree with the reviewers that quantifying this nonlinearity requires more rigorous analysis. To address the concerns about the methodology for calculating temporal hold, we propose an updated method that measures the same phenomenon on a global scale rather than measuring separately for each time point and then averaging. We believe this approach is more reliable and is less prone to unexpected behavior such as local noise or oscillatory behavior as mentioned by the reviewer.

In our updated method, we compare the similarity (correlation) distribution of all DSTRF pairs separated by n  samples when stimulus-aligned (shifting one by n samples), versus when not shifted. This gives us a 1-dimensional vector of distribution difference with increasing pair distance, for which we can use a threshold of significance to select a value for temporal hold.

“Calculation of the temporal hold parameter for a given recording site involves three steps. For a given DSTRF at time *t*, DSTRFt(τ,f), we first calculate its correlation with DSTRFt+n(τ,f) and its shift-corrected version, DSTRFt+n(τ−n,f):(19)αt(n)=corr(DSTRFt(τ,f),DSTRFt+n(τ,f)),1≤n≤30(20)βt(n)=corr(DSTRFt(τ,f),DSTRFt+n(τ−n,f)),1≤n≤30

The upper limit 30 corresponds to 300 ms and was empirically found to be sufficient for the range of temporal hold seen in our data. Next, for each n, we perform a one-tailed Wilcoxon signed-rank test to determine if there is a significant positive change between βt(n) and αt(n) pairs across the entire test set (t=1…T). Finally, the temporal hold is defined as the largest n for which the test yields a significant result (p<0.05). Intuitively, these steps find the largest duration in time for which a spectrotemporal pattern persists, assuming that the latency of that pattern shifts over time to keep it aligned with a specific feature in the stimulus (See Figure 3—figure supplement 1 for examples).”

Subsequently, since our previous definition of shape change was based on the temporal hold calculation method, we have simplified that as well. In our updated method for computing shape change, instead of selecting specific time samples for this calculation, we align all DSTRF samples to their global average, repeating multiple iterations until convergence, and then calculate the sum of normalized singular values as we did before.

“To calculate this parameter, we first removed the effect of temporal hold nonlinearity by time-aligning the DSTRF instances to the average DSTRF across the entire stimuli. This operation is done by finding the best shift, nt, for LLRFs at each time instant, DSTRFt(τ−nt,f). The values of nt are found iteratively by maximizing the correlation between DSTRFt(τ−nt,f) and the average DSTRF over the entire stimulus duration: DSTRF¯(τ,f)=1T∑t=1TDSTRFt(τ,f). At the end of each iteration, the average DSTRF is updated using the new shifted DSTRFs, and this operation is repeated until nt values converge, or a maximum number of iterations is reached. In our data, nt converged within fifty iterations. After removing the temporal hold effect, we repeated the same procedure used for the calculation of network complexity (equation 16) but instead, used the time-aligned DSTRFs to perform the singular-value decomposition. The aligned DSTRFs with same spectrotemporal features but different gain values are captured by the same eigenvectors. The sum of the sorted normalized singular values indicates the diversity of the linear functions learned by the network due to a change in their spectrotemporal feature tuning.”

– Method limitation: If one wants to measure the degree of nonlinearity in a given brain area, isn't it sufficient to simply look at how much of the explainable variance is or is not captured by a linear model? What else is learned by looking at the performance of a particular nonlinear model in this context? In fact, it would seem that using the linear model alone would be the only way to make a general statement in this context. Just because one particular nonlinear model fits one area better than another (or improves predictions in one area more than another) doesn't necessarily show that one or the other area is, in general, more nonlinear. It only suggests the nonlinearity in one area is more readily fit by that particular network given limited training samples. Please comment.

We agree with the reviewers that the unexplained noise-corrected variance by the linear model is in fact indicative of the general nonlinearity of a neural population. Nonlinearity, however, is a very generic description of a transformation and does not really inform us about the types of nonlinear computations that are used. The advantage of our method is that not only it shows improved prediction accuracy over the linear model particularly in higher auditory areas, but more importantly, it unravels three distinct types of nonlinear computation that are present in the neural responses. These computations include gain change, temporal hold nonlinearity, and shape change nonlinearity. How these three different computations contribute to the representation of speech that enables recognition is beyond this paper, but that is a direction that we will pursue in our future work. To clarify this point, we have added the following text to the Discussion:

“We demonstrated how this method can be applied to auditory cortical responses in humans to discover novel types of nonlinear transformation of speech signals, therefore extending our knowledge of the nonlinear cortical computations beyond what can be explained with previous models. While the unexplained noise-corrected variance by a linear model indicates the overall nonlinearity of a neural code, this quantity alone is generic and does not inform the types of nonlinear computations that are being used. In contrast, our proposed method unravels various types of nonlinear computation that are present in the neural responses and provides a qualitative and quantitative account of the underlying nonlinearity.”

– Statistical testing: The statistical testing performed needs to be detailed more to include effect sizes.

We have added the effect values to the text, and also expanded the statistical parts with detail.